

# Effect of seasonality on chemical profile and antifungal activity of essential oil isolated from leaves *Psidium salutare* (Kunth) O. Berg

Delmacia G. de Macêdo[1,*], Marta Maria A. Souza[1,*], Maria Flaviana B. Morais-Braga[1], Henrique Douglas M. Coutinho[2], Antonia Thassya L. dos Santos[2], Rafael P. da Cruz[2], José Galberto M. da Costa[2], Fábio Fernandes G. Rodrigues[2], Lucindo J. Quintans-junior[3], Jackson Roberto G. da Silva Almeida[4] and Irwin Rose A. de Menezes[2]

[1] Department of Biological Sciences, Regional University of Cariri, Crato, Ceara, Brazil
[2] Department of Biological Chemistry, Regional University of Cariri, Crato, Ceará, Brazil
[3] Physiology Department, Federal University of Sergipe, Aracaju, Sergipe, Brazil
[4] Center For Studies and Research of Medicinal Plants, Federal University of San Francisco Valley, Petrolina, Pernambuco, Brazil
[*] These authors contributed equally to this work.

Corresponding authors
Delmacia G. de Macêdo, delmaciamacedo@yahoo.com.br
Irwin Rose A. de Menezes, irwin.alencar@urca.br

## ABSTRACT

Medicinal plants play a crucial role in the search for components that are capable of neutralizing the multiple mechanisms of fungal resistance. *Psidium salutare* (Kunth) O. Berg is a plant native to Brazil used as both food and traditional medicine to treat diseases and symptoms such as stomach ache and diarrhea, whose symptoms could be related to fungal infections from the genus *Candida*. The objective of this study was to investigate the influence of seasonal variability on the chemical composition of the *Psidium salutare* essential oil, its antifungal potential and its effect on the *Candida albicans* morphogenesis. The essential oils were collected in three different seasonal collection periods and isolated by the hydrodistillation process in a modified Clevenger apparatus with identification of the chemical composition determined by gas chromatography coupled to mass spectrometry (GC/MS). The antifungal assays were performed against *Candida* strains through the broth microdilution method to determine the minimum fungicidal concentration (MFC). Fungal growth was assessed by optical density reading and the *Candida albicans* dimorphic effect was evaluated by optical microscopy in microculture chambers. The chemical profile of the essential oils identified 40 substances in the different collection periods with $\gamma$-terpinene being the predominant constituent. The antifungal activity revealed an action against the *C. albicans*, *C. krusei* and *C. tropicalis* strains with an $IC_{50}$ ranging from 345.5 to 2,754.2 $\mu$g/mL and a MFC higher than 1,024 $\mu$g/mL. When combined with essential oils at sub-inhibitory concentrations (MIC/16), fluconazole had its potentiated effect, i.e. a synergistic effect was observed in the combination of fluconazole with *P.salutare* oil against all *Candida* strains; however, for *C. albicans*, its effect was reinforced by the natural product in all the collection periods. The results show that the *Psidium salutare* oil affected the dimorphic transition capacity, significantly reducing the formation of hyphae and pseudohyphae in increasing concentrations. The results show that

*P. salutare* oil exhibits a significant antifungal activity against three Candida species and that it can act in synergy with fluconazole. These results support the notion that this plant may have a potential use in pharmaceutical and preservative products.

## INTRODUCTION

Seasonality variations such as climatic conditions, water restriction, the presence of predators and soil mineral composition may alter secondary plant metabolism (*Figueiredo et al., 2008*) and, consequently, alter the composition of essential oils throughout the year (*Prins, Vieira & Freitas, 2010*). In addition, some specific constituents that present chiral chemical groups are affected by the luminosity rate (*Mulas, Gardner & Craker, 2006*). The isolation of plant essential oils is also influenced beyond taxonomic factors, as well as by the variety of epidermal cellular structures that are responsible for the production and storage of essential oils volatile organic compounds (*Pinto et al., 2007*). Therefore, understanding the seasonal events that alter the quality of the active compounds in the plant is fundamental to support pharmacological studies that contemplate and aim at the formulation of new drugs and direct collection periods in direct commercial plantations of this crop to obtain the oil with greater therapeutic potential.

Many species of the family Myrtaceae have a history of use as traditional medicines in ethnobotanical practices in both tropical and subtropical regions (*Souza et al., 2014*; *Macêdo et al., 2016*). Family members comprise the genera *Eugenia*, *Myrcianthes*, *Campomanesia* and *Psidium*. The *Psidium* genus has approximately 150 species and can be found in all the tropics and subtropics of America and Australia (*Pino et al., 2003*) with several therapeutic potentials already described, especially for *Psidium guajava* Linn. (*Gupta, Chahal & Arora, 2011*; *Joseph & Priya, 2011*). Antimicrobial activity has been described for several species such as *Psidium cattleianum* (*Faleiro et al., 2016*) and *Psidium guineense* (*Fernandes et al., 2012*).

*Psidium salutare* (Kunth) O. Berg., is popularly known in the Northeast region as a "araça preto", often found in Cerrado areas in the Chapada do Araripe, southern Ceará state (*Ribeiro-Silva et al., 2012*), with five varieties of this species being recognized: var. *sericeum*, var. *mucronatum*, var. *decussatum* and var. *pohlianum*, which are also found in other countries such as Paraguay, the Caribbean and Mexico (*Landrum, 2003*). In the Cariri region, in addition to the fruit being edible, the leaves are used in traditional medicine to treat diseases and symptoms such as stomach ache and diarrhea, which may be related to *Candida* infections (*Ribeiro et al., 2014*; *Souza et al., 2014*; *Macêdo et al., 2016*).

Fungal infections caused by dermatophytes and yeasts of *Candida* spp., are a serious health problem in immunocompromised patients in particular and are aggravated by the increase in clinical resistance to the antifungal agents (*Silva et al., 2012*; *Morais-Braga et al., 2016b*). In view of this problem, the interest in the use of vegetable derivatives with

therapeutic potential for antifungal action has intensified (*Macêdo et al., 2015*). These new substances of plant origin may represent alternative and less toxic treatments for the treatment of infections (*Vandeputte, Ferrari & Coste, 2011*), synergism and inhibition of germ tube formation by compounds derived from *Crocus sativus* against *Candida* spp (*Carradori et al., 2016*). Considering the medicinal importance of the genus *Psidium* and the absence of studies with *Psidium salutare*, this is the first study to describe the chemical profile of *P. salutare* leaf essential oil, and the influence of seasonal variation on its composition, antifungal activity and potency to inhibit the morphogenetic switch in *Candida* species.

## MATERIALS AND METHODS

### Collection area of botanical material

*Psidum salutare* leaves were collected in an area of Cerrado *sensu stricto*, at Fazenda Barreiro Grande (latitude: 7°21′41.7″S and longitude 39°28′42.4″W, altitude of 909 m above sea level), located in the Chapada do Araripe, Ceará, Northeast of Brazil, presenting altitudes varying between 870 and 970 m. The region receives on average of 1.043 mm (mm) of rainfall per year (*FUNCEME, 2016*), where they concentrate between January and May with a dry period that lasts between five and seven months, with a critical shortage between July and September (Table 1). According to the Köppen classification system, the climate is hot humid Tropical (Aw) with an average annual of temperature between 24 and 26 °C. The collection is under the authorization of the competent ICMBio with number (no. 50362-2).

### Plant material

Fresh leaves of the species *Psidum salutare* were collected in the months of February, May and August in different periods, dry and rainy season, to evaluate the antifungal activity and the chemical compounds, as described in Table 1, between 8:30 am and 9:30 am. They were then transported to Laboratory of Ecology of Plants of the Regional University of Cariri—URCA. Species exsiccates were produced, identified by Dr. Marcos Sobral (specialist in the Myrtaceae family) and deposited in the Heririum of Caririense Dárdano de Andrade-Lima of the Regional University of Cariri—URCA under number 12601 HCDAL.

### Obtaining and analyzing the essential oil

Approximately 500 g of fresh leaves collected were selected, washed, crushed and submitted to the hydrodistillation process for two hours in a Clevenger type apparatus. The essential oil was then dehydrated with anhydrous sodium sulfate ($Na_2SO_4$) and kept in an amber flask under refrigeration <4 °C until analyzed. The yields were determinate by volume/weight on dry weight basis.

Analysis of the oil was performed using a Shimadzu GC-17 A/MSQP5050A (GC/MS system): DB-5HT capillary column (30 m × 0.251 mm, 0.1 mm of thickness); helium carrier gas at 1.7 mL / min; injection temperature 270 °C; detector temperature 290 °C; column temperature 60 °C (2 min) −180 °C (1 min) at 4 °C/min, then 180–260 °C at

10 °C / min (10 min). The reading speed was 0.5 scan/s of $m/z$ 40–450 with a split ratio of 1:30. The injection volume was 1 μL of 5 mg/mL of ethyl acetate solution. Avoid dead time = 3 min. The mass spectrometer was operated with ionization energy of 70 eV. The identification of the components was by comparison of their respective mass spectrum standards with those registered in the database of the Wiley Online Library and with the calculated retention indices with values in literature (*McLafferty & Stauffer, 2016*; *Adams, 2007*).

## Antifungal activity evaluation
### Culture media and inocula
For the antifungal activity assays, three standard strains of yeast fungi of the genus *Candida* were used: *C. albicans* (CA INCQS 40006) *C. tropicalis* (CT INCQS 40042) and *C. krusei* (CK INCQS 40095) obtained from the Oswald Cruz Cultures Collection of the National Institute of Quality Control in Health (INCQS). All strains were grown on Sabouraud Dextrose agar (SDA-KASVI) and incubated at 37 °C for 24 h. From these, suspensions of the microorganisms were prepared in tubes containing 3 ml of sterile solution (0.9% NaCl). The inoculum concentration was standardized according to the McFarland scale, comparing inoculum turbidity with the 0.5 standard on the scale equivalent to $10^5/10^6$ cells per mL. The potato dextrose agar (PDA, DIFCO) was prepared by diluting it more than that recommended by the manufacturer to make it a depleted medium capable of stimulating yeast to produce hyphae. Agar was added to this diluted medium to obtain a solid medium.

### Determination of the Inhibitory Concentration of 50% of the microorganisms ($IC_{50}$) and obtaining the cellular viability curve
The different *P. salutare* essential oil samples from the periodic collections in the rainy and dry seasons were tested for their antifungal activity. Both the essential oil and antifungal fluconazole (F8929 ≥ 98% (HPLC), powder; Sigma Aldrich, St. Louis, MO, USA) was previously diluted in dimethylsulfoxide (DMSO; Dynamic, Indaiatuba, Brazil) and its final concentration was adjusted with addition of distilled water to obtain the desired concentration for (16,384 μg / ml). The oil and fluconazole solutions were posteriorly microdiluted in Sabouraud Dextrose Broth (SDB) medium in a serial concentration manner ranging from 8,192 to 8 μg/mL in 96-well plates. The penultimate well, the latter serving as a growth control (*Javadpour et al., 1996*). The concentration of DMSO at the oil concentrations ranged from 5 to 0.004%. Product dilutions (using saline instead of inoculum) and medium sterility controls were also achieved. The plates were then taken to an incubation chamber for 24 h at 37 °C and following this period the plates were read using an ELISA spectrophotometer (Thermoplate®) apparatus. The results obtained in the ELISA reading were used to construct the cell viability curve and to determine the $IC_{50}$ of the *P. salutare* essential oils (*Morais-Braga et al., 2016b*). All test were perfomed in triplicate.

**Table 1 The average annual of meteorological conditions for each collection (2016).**

| Collection period | February | May | August | 2016 Average annual |
|---|---|---|---|---|
| | OEFPs1/ winter | OEFPs2/ winter | OEFPs3/ summer | |
| Precipitation (mm) | 49 | 145 | 0.0 | 968.0 |
| Yield (%) | 0.73 | 0.29 | 0.15 | |
| Temperature (°C) | 26 | 27 | 35 | |

**Notes.**

OEFPs, essential oil of *Psidium salutare* sheets, 1, 2, 3 collection.

### *Determination of the minimal fungicidal concentration (MFC)*

A small sterile rod was placed in each well of the microdilution test plate, with the exception of the sterility control. After mixing the medium in each well, the rod was taken to a large petri dish containing SDA, where by touching the surface, the solution (medium + inoculum + natural product) was transferred for yeast subculture and cell viability analysis. The plates were incubated at 37 °C for 24 h, and checked for the growth or non-growth of *Candida* colonies (*Ernst et al., 1999*). The concentration at which there was no growth of fungal colonies was considered the MFC of the natural product.

### Evaluation of the *Psidium salutare* essential oil modulating effect on the antifungal activity of fluconazole

The solution containing the essential oil of *P. salutare* (OEFPs) was tested in subinhibitory concentration (MFC/16). The volume of 100 μL of a solution containing SDB (Sabouraud Dextrose Broth), 10% inoculum and natural product were distributed in each well in the alphabetical direction of the plate. Afterwards, 100 μL of the antifungal were mixed to the first well and serially microdiluted in a ratio of 1:1, the latter cavity being used as fungus growth control (*Coutinho et al., 2008*). The fluconazole concentrations varied gradually from 8,192 to 8 μg/mL. Dilution controls of the natural products (OEFPs) were used where the inoculum was replaced by saline/DMSO and control of sterility with the medium. The plates were incubated at 37 °C for 24 h and reading was done on a spectrophotometer, Thermoplate® ELISA, with a wavelength of 630 nm (*Morais-Braga et al., 2016b*).

### Effect of the *Psidium salutare* leaf oil on *Candida albicans* morphogenesis

The essential oil from the three samples collected at different periods were used to observe if the natural product caused any alteration in the morphogenesis of *C. albicans*, the oil was tested in different concentrations such as the Superior Evaluated Concentration: SEC (8,192 μg/mL), SEC/4 (2,048 μg/mL) and SEC/16 (512 μg/mL).

The trials were performed with some modifications according to *Sidrim & Rocha (2003)* and *Mendes Giannini et al. (2013)*. The medium (3 mL) were combined to the tested product, were poured into the slide of the microscope at the respective concentrations, previously homogenized by the agitator. After solidification of the medium, the yeast was seeded with the aid of a 1 μL calibrated loop and two parallel grooves were extracted. The striae were covered with sterile lamellae. Plates were incubated and after 24 h as slides were subsequently observed under a 40× objective optical microscope. A control for yeast
**Table 2** Determination of the percentage composition of the chemical composition of the *Psidium salutare* leaf essential oil by gas chromatography coupled to mass spectrometry (CG/MS) in different collection periods.

| Compounds | tR* (min) | OEFPs1 | OEFPs2 | OEFPs3 | % (media) |
|---|---|---|---|---|---|
| 1,8 Cineole | 5.5 | 0.61[a] | 0.51[a] | 1.05[a] | 0.72 |
| Dimethyl benzylcarbinyl acetate | 8.2 | 0.19[a] | 0.15[a] | 0.65[a] | 0.33 |
| Copaene | 11.2 | 3.22[a] | 3.53[a] | 1.91[a] | 2.89 |
| Cubenol | 14.8 | 0.63[a] | 0.0[a] | 3.42[c] | 1.35 |
| Espatulenol | 12.2 | 0.23[a] | 0.30[a] | 0.13[a] | 0.22 |
| Sabinene hydrate | 8.1 | 2.28[a] | 3.48[a] | 3.94[a] | 3.23 |
| Isocarofilene | 11.9 | 3.78[a] | 3.75[a] | 1.20[a] | 2.91 |
| Limonene | 5.4 | 1.10[a] | 1.15[a] | 1.14[a] | 1.13 |
| Linalool | 6.6 | 5.55[a] | 4.72[a] | 7.26[a] | 5.84 |
| Myrcene | 4.7 | 0.65[a] | 0.42[a] | 0.08[b] | 0.38 |
| Myrtenol | 7.5 | 0.21[a] | 0.16[a] | 0.09[a] | 0.15 |
| Ocimene | 5.7 | 2.15[a] | 1.93[a] | 1.50[a] | 1.86 |
| Palustrol | 14.1 | 0.05[a] | 0.11[a] | 0.10[a] | 0.09 |
| Patchoulane | 14.7 | 0.25[a] | 0.19[a] | 3.08[a] | 1.17 |
| P-Cymene | 5.3 | 5.05[b] | 6.37[b] | 17.83[e] | 9.75 |
| Selina-3,7 (11) -diene | 13.6 | 0.37[a] | 0.28[a] | 0.0[a] | 0.22 |
| Seychellene | 13.9 | 0.20[a] | 0.17[a] | 0.40[a] | 0.26 |
| Terpineol | 8.3 | 1.67[a] | 0.90[a] | 0.12[a] | 0.90 |
| Terpinolene | 6.4 | 16.99[c] | 14.49[c] | 6.90[b] | 12.79 |
| Valencene | 13.3 | 0.23[a] | 0.09[a] | 0.30[a] | 0.21 |
| Viridiflorene | 16.4 | 0.12[a] | 0.0[a] | 0.35[a] | 0.16 |
| Viridiflorol | 14.7 | 0.53[a] | 0.95[a] | 2.07[a] | 1.18 |
| α-phellandrene | 5.0 | 0.15[a] | 0.08[a] | 0.05[a] | 0.09 |
| α-caryophyllene | 12.4 | 0.24[a] | 0.29[a] | 1.68[a] | 0.74 |
| α-cubebene | 15.0 | 0.90[a] | 2.05[a] | 0.0[a] | 0.98 |
| α-farnesene | 13.8 | 0.03[a] | 0.03[a] | 0.24[a] | 0.10 |
| α-gurjunene | 11.7 | 0.21[a] | 0.10[a] | 0.08[a] | 0.13 |
| α-muurolene | 12.6 | 0.63[a] | 0.72[a] | 0.69[a] | 0.68 |
| α-pinene | 5.1 | 0.83[a] | 0.55[a] | 0.62[a] | 0.67 |
| β-cadinene | 5.9 | 0.96[a] | 1.45[a] | 0.83[a] | 1.08 |
| β-elemene | 13.7 | 0.16[a] | 0.12[a] | 0.70[a] | 0.33 |
| β-eudesmene | 12.8 | 0.15[a] | 0.11[a] | 0.0[a] | 0.09 |
| β-guaienum | 15.5 | 2.79[a] | 3.12[a] | 0.0[a] | 1.97 |
| γ-gurjunene | 13.6 | 0.10[a] | 0.21[a] | 0.26[a] | 0.19 |
| γ-muurolene | 13.2 | 2.58[a] | 2.42[a] | 3.20[a] | 2.73 |
| γ-terpinene | 5.9 | 13.97[d] | 17.09[d] | 10.32[c] | 13.79 |
| δ-cadinene | 13.3 | 5.27[a] | 3.88[a] | 3.84[a] | 4.33 |
| δ-cadinol | 15.3 | 1.68[a] | 0.0[a] | 0.92[a] | 0.87 |
| δ-guaiene | 14.9 | 0.28[a] | 0.28[a] | 3.70[b] | 1.42 |
| τ-cadinol | 15.2 | 12.75[d] | 10.51[d] | 10.35[d] | 11.20 |

**Table 2** (*continued*)

| Compounds | tR* (min) | OEFPs1 | OEFPs2 | OEFPs3 | % (media) |
|---|---|---|---|---|---|
| Monotherpenes hydrocarbons | | 40.06 | 41.53 | 37.82 | 39.79 |
| Sesquiterpenes hydrocarbons | | 21.65 | 22.15 | 18.98 | 20.94 |
| Oxygenated monotherpenes | | 11.15 | 10.32 | 13.08 | 12.98 |
| Oxygenated sesquiterpenes | | 15.87 | 11.87 | 16.99 | 14.91 |
| Others | | 1.01 | 0.79 | 4.13 | 1.44 |
| Total | | 89.74 | 86.66 | 91.00 | 90.6 |

**Notes.**

TR, retention time; OEFPs, essential oil from the leaves of *Psidium salutare*; first collection (February), second collection (May), third collection (August). Averages followed by different letters differ by Tukey test at $p < 0.05$.

growth (hyphae stimulated by depleting medium) was performed, as well as a control with the conventional antifungal fluconazole for comparative purposes. Tests using DMSO as a control were previously performed (*Morais-Braga et al., 2016a*), demonstrating that it does not cause inhibition of hyphae at the concentrations tested.

## Statistical analysis

The data obtained for each sample were checked for their normal distribution and then analyzed by one-way ANOVA followed by Tukey's test. The $IC_{50}$ values were computed by linear regression for interpolation in standard curves relating the percentage (%) growth values and the concentration of the product in µg/mL using the GraphPad Prism software, version 6.0. All analyzes were performed in triplicates (see raw data in the Supplementary Information attached).

## RESULTS

In the evaluation of the yield of the essential oil *Psidium salutare* in the analyzed periods, February (0.73%), May (0.29%) and August (0.15%) show that highest yields coincident with the precipitation periods and dryness in the region, however, it was not possible to obtain a statistically significant correlation. Then, when analyzing the *P. salutare* oil yield, the beginning of the rainy season was the ideal period for collection. In the *P. salutare* GC/MS analysis it was possible to identify an average of 89.13% of the constituents corresponding to 40 compounds (Table 2). When calculating the average of the compounds, a predominance of monoterpene hydrocarbons (39.79%), sesquiterpene hydrocarbons (20.94%), oxygenated monoterpenes (12.98%) and oxygenated sesquiterpenes (14.91%) can be observed.

The major constituents were linalool, p-Cymene, terpinolene, γ-terpinene and $\tau$-cadinol. The results obtained during the collection periods showed that although several compounds presented a random composition, others remained constant. In the rainy season in February and May, the compounds that stood out were terpinolene (14.49 - 16.99%), γ-terpinene (13.97–17.09%), $\tau$-cadinol (12.75–10.51%), p-Cymene (5.05–6.37%) and linalool (5.55–4.72%). In the dry season, represented by August, the major compounds were p-cymene (17.83%), γ-terpinene (10.32%), $\tau$-cadinol (10.35%) and linalool (7.26%). In the dry season, represented by August, the major compounds were p-cymene (17.83%), γ-terpinene (10.32%), $\tau$-cadinol (10.35%) and linalool (7.26%). This

**Table 3** The inhibitory effect of association the essential oil of *P. salutare* with fluconazole on Candida ($\mu$g/mL).

| Tested Products | Strains | | | | | |
|---|---|---|---|---|---|---|
| | CA INCQS 40006 | | CT INCQS 40042 | | CK INCQS 40095 | |
| | CFM $\mu$g/mL | IC$_{50}$ $\mu$g/mL | CFM $\mu$g/mL | IC$_{50}$ $\mu$g/mL | CFM $\mu$g/mL | IC$_{50}$ $\mu$g/mL |
| Fluconazole (FCZ) | 8,192 | 16.8 | $\geq$16,384 | 9.3 | $\geq$16,384 | 271 |
| OEFPs 1+FCZ | 1,024 | 2.7 | $\geq$16,384 | 2.6 | 8,192 | 44.4 |
| OEFPs 2+FCZ | 8,192 | 8.0 | $\geq$16,384 | 5.3 | 1,024 | 32.4 |
| OEFPs 3+FCZ | 4,096 | 6.3 | $\geq$16,384 | 3.7 | 8,192 | 45.2 |

**Notes.**

OEFPs, essential oil of *Psidium salutare* leaves, 1, 2 and 3 collections; CA, *Candida albicans*; CT, *Candida tropicalis*; CK, *Candida krusei*; INCQS, National Institute of Health Quality Control; IC$_{50}$ ($\mu$g/mL), the inhibitor concentration that decreases 50% of the growth.

variation can be partly explained by the fact that environmental factors can affect certain chemical compounds while exerting any influence on the production of other chemicals.

The intrinsic *P. salutare* essential oil antifungal activity at different collection times, against different *Candida* strains showed no significant clinical activity (MIC $\geq$ 1,024 $\mu$g/mL), demonstrating that it was little influenced by changes in the chemical composition of the oil and by rainfall (Table 3). In this sense, although punctually significant, the chemical variations in the oil composition were not able to exhibit satisfactory inhibitory effect against Candida strains showing effects only in high concentrations, *Candida albicans* INCQS 40006 (4,096 $\mu$g/mL), *Candida tropicalis* INCQS 40042 ($\geq$16,384 $\mu$g/ml) and *Candida krusei* INCQS 40095 (1,024 $\mu$g/ml) (Table 3).

Among the analyzed periods, the IC$_{50}$ (Ability to Inhibit 50% of cells), of products ranged from 345.5 to 2,754.2 $\mu$g/mL and image of the cellular viability curve in different concentrations of essential oil, the lowest value recorded for *C. albicans* was related to the first collection period, with an IC$_{50}$ of 581.3 $\mu$g/mL (Fig. 1A), precipitation of 49 mm and elevated major compounds, such as terpinolene, $\tau$-cadinol and $\gamma$-terpinene. For *C. tropicalis*, the lowest IC$_{50}$ value was observed in the last collection period (1,621.8 $\mu$g/mL) (Fig. 1C), coinciding with the dry period in the region, with p-cymene, linalol, $\gamma$-terpinene and $\tau$-cadinol in higher concentrations in the sample.

However, for *C. krusei* the antifungal activity was showed more active in second collection period (345.5 $\mu$g/mL) (Fig. 1E) with significant value when compared to fluconazole IC$_{50}$ of 271.3 $\mu$g/mL (Table 4). This result corroborates with an incidence of precipitation of 145 mm and presence of the major $\gamma$-terpinene, $\tau$-cadinol and terpinolene compounds in the sample.

For the Intrinsic Minimal Fungicide Concentration (MFC) the results showed a chemical variation in the essential oil composition between dry and rainy periods, thus influencing the concentration for *C. albicans* (4,096 $\mu$g/mL) and *C. krusei* (1,024 $\mu$g/mL), however for *C. tropicalis* ($\geq$16,384 $\mu$g/mL) the concentrations remained constant.

In the verification of the potential modifier of the effect of fluconazole by the essential oil (Table 4), we can verify that there was a modulatory activity for all strains (Figs. 1B, 1D, 1F), especially for *C. albicans* 40006 (2.7 to 8.0 $\mu$g/mL), which exhibited lower concentrations than when compared to fluconazole alone (IC$_{50}$ 16.7 $\mu$g/mL), exhibiting

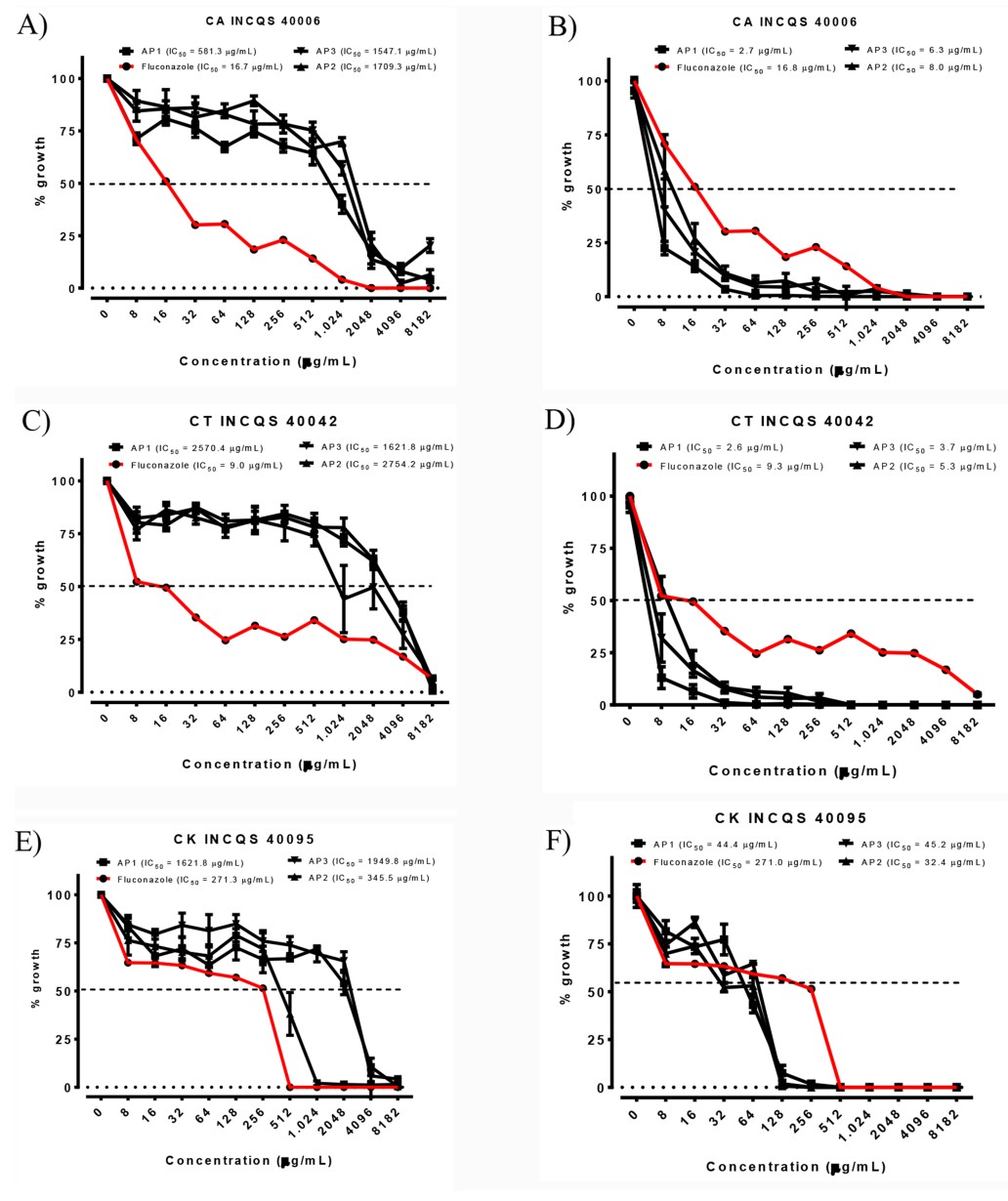

**Figure 1** **Cell viability curve and IC$_{50}$ of the *P. salutare* essential oil (A, C and E) and the oil in combined with fluconazole (B, D and F) against different *Candida* spp. strains, at different collection periods.** Concentration of fluconazole: 2,048 µg/mL. OEFPs, Essential oil of the leaves of *Psidium salutare*, 1, 2 and 3 collections; CA, *C. albicans*; CT, *C. tropicalis*; CK, *C. krusei*; INCQS, National Institute of Quality Control in Health. (A) Cell viability curve and IC$_{50}$ of *Psidium salutare* essential oil against *Candida albicans*. (B) Cell viability curve and IC$_{50}$ of *Psidium salutare* essential oil combined with fluconazole against *Candida albicans*. (C) Cell viability curve and IC$_{50}$ of *Psidium salutare* essential oil against *Candida tropicalis*. (D) Cell viability curve and IC$_{50}$ of *Psidium salutare* essential oil combined with fluconazole against *Candida tropicalis*. (E) Cell viability curve and IC$_{50}$ of *Psidium salutare* essential oil against *Candida krusei*. (F) Cell viability curve and IC$_{50}$ of *Psidium salutare* essential oil combined with fluconazole against *Candida krusei*.

**Table 4** The CFM (μg/mL) of the essential oil of *P. salutare* on different strains of Candida in modulatory effect.

| Strains | Tested Products | | | | | |
| --- | --- | --- | --- | --- | --- | --- |
| | OEFPs1 μg/mL | OEFPs1+FCZ μg/mL | OEFPs2 μg/mL | OEFPs2+FCZ μg/mL | OEFPs3 μg/mL | OEFPs3+FCZ μg/mL |
| CA INCQS 40006 | 1,024 | 581.3 | 8,192 | 1,709.3 | 4,096 | 1,547.1 |
| CT INCQS 40042 | ≥16,384 | 2,570.4 | ≥16,384 | 2,754.2 | ≥16,384 | 1,621.8 |
| CK INCQS 40095 | 8,192 | 1,621.8 | 1,024 | 345.5 | 8,192 | 1,949.8 |

**Notes.**
OEFPs, essential oil of leaves of *P. salutare*, 1,2 and 3 collections; CA, *C. albicans*; CT, *C. tropicalis*; CK, *C. krusei*; INCQS, National Institute of Quality Control in Health.

synergism in all curves at all collection periods analyzed, promoting an inhibitory effect on microorganisms, greater than sum of the effects of individuals.

The effect of the natural product on morphological transition in *C. albicans* was evaluated by microcultive assay. It can be observed that the essential oil inhibited the formation of hyphae and pseudohyphae at concentrations starting from 512.0 μg/mL, resulting in the reduction of fungal virulence (Fig. 2). The microscopy results shown in Fig. 2 demonstrate that the essential oil can inhibit germinative tube formation and reduce hyphae elongation, which can be considered effective against *C. albicans* dimorphism, thus reducing fungal progression and the spread of infection.

## DISCUSSION

As expected, the chemical composition varied during the analyzed period. This result corroborates with other studies that have shown that environmental factors can affect certain chemical compounds, while in others they have no influence on their production (*Araújo et al, 2015*; *Estell, Fredrickson & James, 2016*). In the leaves of *Camellia sinensis*, the main catechins (epigallocatechin gallate, epicatechin) varied during the year, and this variation was associated to the following environmental factors that can act in combination or alone: day length, sunlight and / or temperature (*Yao et al., 2016*). According to studies the rainy season was favorable for the production of *Copaifera langsdorffii* Desff oils (*Souza de Oliveira et al., 2017*), *Eucalyptus citriodora* Hook (*Castro et al., 2008*) and *Cymbopogon citratus* (*Santos et al., 2009*).

In the present study, the production of the major compounds such as Terpinolene and γ-terpinene in the milder months (26 °C) was positively influenced, with an increase in the percentage of these compounds in the sample, whereas the production of P-cymene was positively influenced in the warmer period, which shows that similar compounds can be altered simultaneously by the same factor, resulting in variations throughout the year, except for the τ-cadinol compound that remained stable.

Comparing the results with available literature, the rainy season was also favorable for the yield of oils *Copaifera langsdorffii* Desff (*Souza de Oliveira et al., 2017*), *Eucalyptus citriodora* Hook (*Castro et al., 2008*) and *Cymbopogon citratus* (*Santos et al., 2009*). The presence of the major compounds as, γ-terpinene, terpinolene, τ-cadinol, p-cymene and linalool in the essential oil of the species under study, were also present in *P. myrsinites*

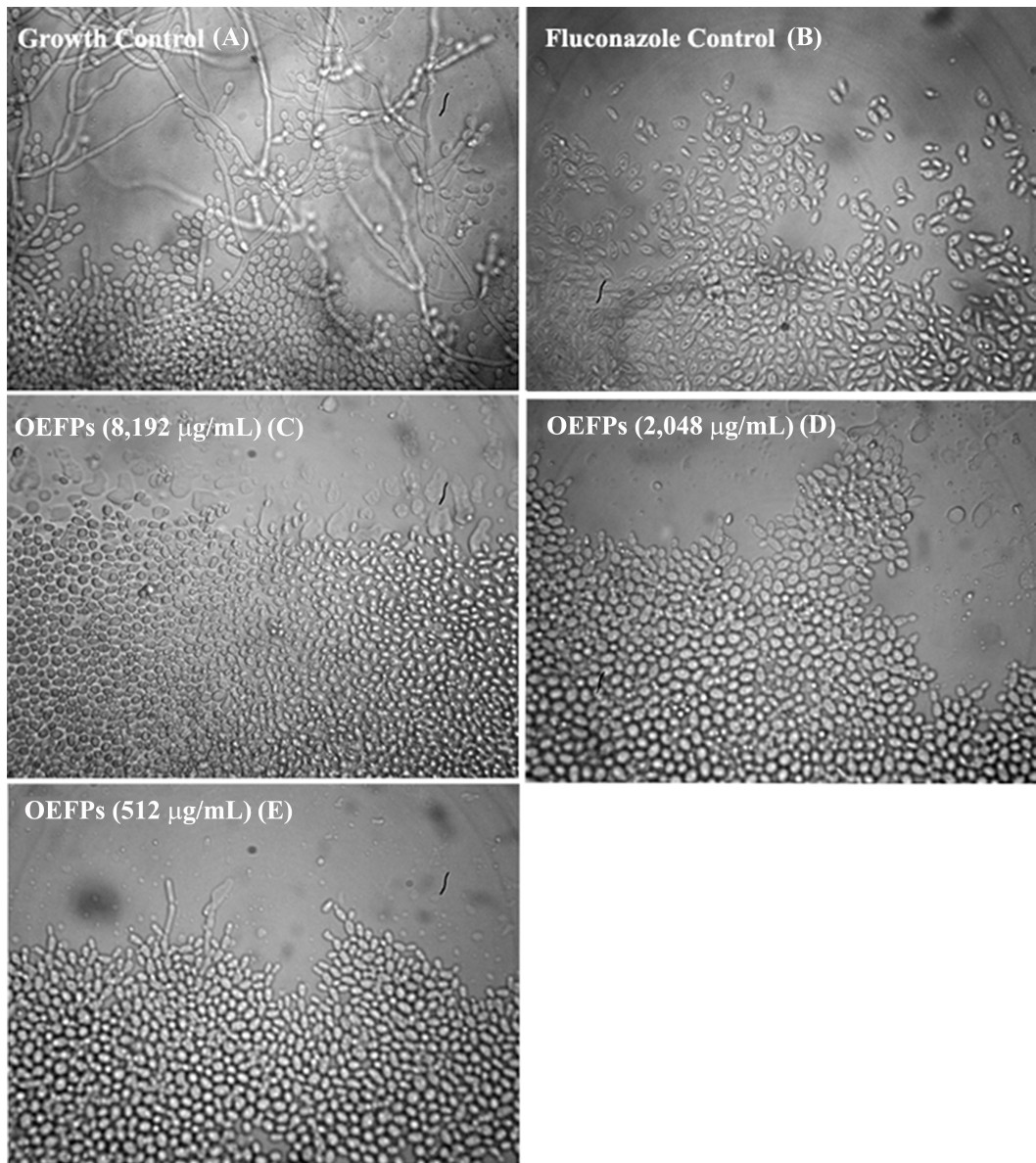

**Figure 2** **Effect of the *Psidium salutare* essential oil on *Candida albicans* yeast micromorphological aspects.** Culture performed in depleted potato dextrose agar medium, with 40× objective visualization. (A) Growth control, (B) fluconazole antifungal effect at 2,048 μg/mL, (C) *P. salutare* essential oil effect at 8,192 μg/mL, (D) *P. salutare* essential oil effect at 2,048 μg/mL and (E) (C) *P. salutare* essential oil effect at 512 μg/mL; CA, *C. albicans*; INCQS, National Institute of Quality Control in Health.

Mart. (*Medeiros et al., 2015*), *Psidium pohlianum* O. Berg, *Psidium guyanensis* Pers (*Neto et al., 1994*) and *Psidium caudatum* McVaugh (*Yáñez et al., 2002*).

Variations in plant active components are important parameters to correlate biological activity, including antibacterial, antifungal and insecticide. Knowledge of the abiotic factors influencing the chemical variability and essential oil yield is important for optimizing crop conditions and harvesting time so that they are of high quality, factors essential

for commercialization. In addition, a number of biotic factors such as plant/micro-organism (*Stoppacher et al., 2010*), plant/insects (*Kessler & Baldwin, 2001*) and plant/plant interactions, age and development stage, as well as abiotic factors such as luminosity (*Takshak & Agrawal, 2016*), temperature, rainfall, nutrition, time and harvest time (*Bitu et al., 2015*), may present correlations with each other, acting in conjunction, and may exert a joint influence on the chemical variability and essential oil yield.

The major compounds terpinolene, $\tau$-cadinol and y-terpinene, have already been reported in other plants that have been studied for their antifungal activity against *C. albicans* (*Tampieri et al., 2005*); however, none of the compounds were studied to evaluate their activity against *C. krusei*. Moreover, studies with *C. tropicalis* verify that the compounds p-cymene and linalool also possess an inhibitory effect (*Hsu et al., 2013*; *De Oliveira Lima et al., 2017*).

Combination therapy using natural and antimicrobial products has been reported as an important strategy to combat the development of microbial resistance due to the production of an additive or synergistic effect. Thus, we demonstrated that the essential oil association with fluconazole may represent a therapeutic benefit in reducing the antifungal dosage, representing an improvement in toxic levels, while producing a fungicidal effect (*Pemmaraju et al., 2013*).

This is the first work that shows the potential modulating activity of *P. salutare* essential oil, so there was no way to compare the results here with respect to the species. Within the genus *Psidium*, some data exist; however, they are related to extracts. *Morais-Braga et al. (2016a)* and *Morais-Braga et al. (2016b)* observed fluconazole and all extracts had high inhibitor concentrations, however, when these were in association with sub-inhibitory concentrations (MIC/16), fluconazole had an improved effect, thus a synergistic effect was observed in the combination of fluconazole with extracts of *Psidium brownianum* against all strains of *Candida* (*Morais-Braga et al., 2016b*). According to this study, *Castro et al. (2015)* demonstrated that the *P. cattleianum* essential oil had an effect on the inhibition of important clinical fungal strains such as *Trichosporon asahii* (*Castro et al., 2015*), *C. parapsilosis*, *C. albicans*, *C. lipolytica* and *C. guilhermondi*, with concentrations ranging from 41.67 ± 18.04 to 16,670 ± 72.17 µg/mL for the tested strains.

*C. albicans* was selected for association study because it is the most common pathogenic agent involved in systemic infections and the main strain responsible for infections caused by *Candida* fungi (*Romani, 2012*; *Yapar, 2014*). In the previous studies, of our research group, with other species of this genus, *P. brownianum* and *P. guajava* extracts had their antifungal potential investigated, obtaining favorable results, where they also managed to affect the phenotypic plasticity of *C. albicans* and *C. tropicalis*, reducing the formation process of hyphae and pseudohyphae as their concentrations were increased (*Morais-Braga et al., 2016b*; *Morais-Braga et al., 2016a*; *Morais-Braga et al., 2017*).

*C. albicans* is a polymorphic fungus that can grow both in the yeast form (ovoid form), elongated ellipsoid cells with constrictions in the septa (pseudohyphae), or as true hyphae of parallel walls, as can be observed in Fig. 2A. The hyphae or pseudohyphae forms are responsible for the infectious process ranging from superficial skin infections to life-threatening systemic infections. Transition to the hyphae form can be triggered by

increases in temperature to 37 °C, increases in pH, and the addition of inducers (*Shareck & Belhumeur, 2011*). In this form, the germinating tube and the tip extension can generate strong pressures for tissue penetration due to the secretion of proteases, lipases and other histological enzymes. This is important since hyphae formation is central to another aspect of *Candida*'s virulence: development of biofilms that is associated with increased resistance to antifungal medications (*Calderone & Fonzi, 2001*).

Some authors have proposed that the essential oil activity may be in part related to its hydrophobicity, responsible for its partition of the cell membrane lipid bilayer, leading to a change in permeability and cell membrane damage resulting from direct damage to the membrane resulting in a reduced ability to maintain cellular functions (*Braga et al., 2007*; *Hsu et al., 2013*). Another mechanism is related to a metabolic impairment with a reduction of $3':5''$-cyclic adenosine monophosphate (cAMP) formation and, together with a mitogenic activation protein (MAP) signaling pathway, responsible for playing an important role in the formation of filamentous forms (*Hollosy & Keri, 2004*; *Deveau et al., 2010*; *Dižová & Bujdáková, 2017*). Several mechanisms have been tested in order to provide new and valuable means to combat *Candida* pathogenesis that may lead to new strategies for the development of antifungal drugs.

## CONCLUSION

In conclusion, the essential oil of *P. salutare* presented, as its main components, hydrogenated monoterpenes and γ-terpinene, whose composition was influenced by the beginning of the rainy season, proving this to be the ideal period for the isolation of the oil. It is not possible to affirm that the antifungal activity of the oil was influenced by the seasonal changes in the precipitation, with the exception of *C. krusei*, where it presented the lower MFC and $IC_{50}$ values. The essential oil demonstrated a significant effect on Candida morphogenesis, reducing the ability of morphological transitions from invasive infectious processes and resistance to *C. albicans*. In this way, the presented results can be a starting point for new in vivo assays for the possible development of new complementary and alternative therapies, as well as to support its popular medicinal use against diseases of fungal origin.

### Funding

The authors received funding for this work by CAPES, FUNCAP, CNPq and FINEP. The funders had no role in study design, data collection and analysis, decision to publish, or preparation of the manuscript.

### Grant Disclosures

The following grant information was disclosed by the authors:
CAPES.
FUNCAP.

CNPq.
FINEP.

## Competing Interests

The authors declare there are no competing interests.

## Author Contributions

- Delmacia G. de Macêdo and Antonia Thassya L. dos Santos performed the experiments, prepared figures and/or tables, approved the final draft.
- Marta Maria A. Souza, Henrique Douglas M. Coutinho and Irwin Rose A. de Menezes conceived and designed the experiments, analyzed the data, approved the final draft.
- Maria Flaviana B. Morais-Braga conceived and designed the experiments, performed the experiments, prepared figures and/or tables, approved the final draft.
- Rafael P. da Cruz approved the final draft.
- José Galberto M. da Costa conceived and designed the experiments, contributed reagents/materials/analysis tools, approved the final draft.
- Fábio Fernandes G. Rodrigues performed the experiments, approved the final draft.
- Lucindo J. Quintans-junior and Jackson Roberto G. da Silva Almeida analyzed the data, authored or reviewed drafts of the paper, approved the final draft, help in the correction of the manuscript.

## Field Study Permissions

The following information was supplied relating to field study approvals (i.e., approving body and any reference numbers):

The collection is under the authorization of the competent ICMBio with number (no 50362-2).

## Data Availability

The raw data are provided in Data S1.

## Supplemental Information

Supplemental information for this article can be found online at http://dx.doi.org/10.7717/peerj.5476#supplemental-information.

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
