# Peer review of "Effect of seasonality on chemical profile and antifungal activity of essential oil isolated from leaves Psidium salutare (Kunth) O. Berg"

_PeerJ, doi:10.7717/peerj.5476_

## Round 0.1 · original submission · Major Revisions

Your manuscript has been reviewed by several experts and 2 have raised concerns about the antifungal assays you performed. Please address their comments in full in your revised manuscript.

Reviewer 1 ·

Basic reporting

The manuscript describes the seasonal variations of Psidum salutare essential oils (yield and composition) and the antifungal activity of these.
The manuscript is well organized and well structured. It meets the expectations and standards of PeerJ. The subject discussed is very interesting because few studies have examined the seasonal effect of essential oils in this species. This is therefore an originality that is well emphasized in the introduction.

Experimental design

The experimental protocol is well done with approved and proven methods.
The only lack in this part is related to the lack of information on essential oil used for antifungal tests. What stage of development?

Validity of the findings

The results obtained are very interesting and can make it possible to have a modulation of the composition of the essential oils as a function of the stage of development and or seasons.

Additional comments

The manuscript suffers from several gaps
1- For a better follow-up of the different phenological stages, it is advisable to follow a standard scale of the species, if any, of a species of the same botanical family or by the use of the BBCH scale.
2- it is not specified the essential oil of which season is used for the antifungal tests. This is one of the weakest manuscript points. Since the title of the article is the seasonality of the composition of the essential oils of antifungal activity. However, if the seasonality of the composition is present, that of the antifungal activity is not.

The other weak point concerns the climatic conditions. many parameters are missing, including temperatures. It is now well known that essential oils are produced and modulated according to climatic conditions, in particular in response to biotic and abiotic stresses. For these, there is not only water but especially temperatures. The authors have completely forgotten this parameter. Moreover, the variation of essential oils and the impact of climatic conditions on this variation is not well discussed

·

Basic reporting

state novelty

Experimental design

fine

Validity of the findings

state reason for selecting the plant

Additional comments

All the experiments are well performed. The article can be accepted after stating its novelty in the abstract and conclusion

Reviewer 3 ·

Basic reporting

The manuscript is poorly written which makes the understanding of the data and interpretation difficult. I have highlighted sentences and example or where it must be re-phrased. The figure legends are very basic and incorrect if Figure 2 it seems (concentration units are not correct). The legend of the graph in Figure 1 are too small to read.

Experimental design

It is apparent that the authors have expertise in plants and essential oils. The concept of seasonality is relevant to plant extracts.

The antifungal assays however are more difficult to follow.

The method section needs extensive re-writing to adapt to standard procedures commonly used in the field of antifungals. It is not easy to reproduce the assays as they are described in the manuscript as essential details are missing.

Assays were performed three times according to the statistic section, which is good.

Validity of the findings

Conclusions should remain specific to the conditions used in the in vitro assays, as the choice of the conditions are atypical in the field. Assays that are typically used are recommended to the authors so their results can be extrapolated a bit further.
The drug test are statistically good and reliable.
The determination of the effect of the samples on Candida morphogenesis is less convincing. It is not clear how many times the assays were done. The concentration units appear to be wrong on the figure itself. Other hyphal inducers are typically used but they are not mentioned in the current manuscript.
The authors refer to fungicidal activity through out the text, yet the essential oil samples do not appear to be cidal even at the highest concentrations. Re-phrasing is required to make sure the appropriate wording is used. Recommendations were made to the authors in the General comments to authors section.

Additional comments

The manuscript aims to study the effect of seasonality on the composition and antifungal activity of plant material. The concept is relevant and attractive.

In abstract:
Line 29: “….effect on the Candida albicans virulence micromorphology”. Do the author mean the effect on Candida albicans morphogenesis (also known as the morphogenetic switch or dimorphic transition)? “Virulence micromorphology” should be replaced through the whole text by morphogenesis or morphogenetic switch.
Line 33: “through” should be replaced by “using the broth microdilution assay”
Line 34: “fungicidal” should be replaced by “inhibitory”, MIC instead of MFC, as it is not clear how the authors have demonstrated whether the essential oil kills the fungal cells or inhibits their growth. Killing and fungicidal activity would require CFUs analyses to demonstrate whether cells are alive and not growing, or dead.
Line 35: “…dimorphemic effect”..should be replaced by the “dimorphic switch/transition”
Line 38: “The antifungal activity revealed an action against..” should be replaced by “MIC assays revealed an antifungal activity against…”
Lines 39 to 41: The whole sentence starting from “in the antifungal modulating…” needs to be re-phrased. What is the antifungal modulating assays? Was the activity significant? Was this tested with statistics?
Lines 41 to 44: those sentences should be at the present tense: “The results show that…
Line 44: a sentence like this “The results show that P. salutare oil exhibits a significant antifungal activity against three Candida species and that it can act in synergy with fluconazole” would be clearer. What is a fluconazole potentiator?

Line 48 and Keywords: I would remove precipitation, which is included in seasonal variation; pathogenesis and micromorphology could be replaced by Candida species and morphogenesis respectively.

In the introduction:
Line 58: remove “In this sense”
Lines 61-63: do the authors mean that the knowledge of the impact of seasonality on the essential oil composition can help identifying highly functionalized substances, which may have a greater commercial relevance?
Line 81: remove “especially”
Lines 80-82: Fungal infections caused….are a serious health problem in immunocompromised patients in particular and is aggravated by the increased resistance of clinical isolates to antifungals.
Lines 84 to 86: These derivatives of plant origin may represent alternative and less toxic treatments against …and the sentence that follows is incomplete, Lines 86 to 88.
Line 88: remove “In this sense”
Lines 88 to 91: “…..to describe the chemical profile of P. salutare leaf essential oil, and the influence of seasonal variation on its composition, antifungal activity and potency to inhibit the morphogenetic switch in Candida species.

In Material & Methods:
Lines 100-103: the sentence is missing a verb

In section Culture media and inocula: cells can be grown ON Sabouraud Dextrose agar, or IN Sabouraud dextrose liquid broth, but they cannot be grown IN Sabouraud Agar.

In Section 2.3.2: this section needs re-written as important clues are missing. The final cell density in the assay has to be indicated, the final concentration of DMSO in each wells must be indicated. Was DMSO used at 0.5% final, 1%, etc?. The way the authors chose to have their cells in eppendorfs or other recipients is not relevant here. Final concentrations, final dilutions, which temperature was used, for how long, etc. Is it IN sabouraud dextrose liquid medium? Or is it ON sabouraud agar?
Line 156: what is CDD?
In addition, typically one starts with measuring MIC50 and not MFC50 as it indicates growth inhibition as a first step. Whether a compound is cidal of static is determined by a different type of assay (refer to killing assays).

In section 2.3.3: This method is atypical. CFU counts by mean of killing assay over time is typically used for this kind of determination. Check killing assay for Candida species in published work.

In section 2.4:
What is CSD? What is the final cell density? What is the final fluconazole concentration? What is the natural product? Final concentration of DMSO? What temperature? For how long?
Use simpler writing to indicate important clues on how the assay was ran so someone else can reproduce it.

In section 2.5:
What is SEC? the first sentence is missing a verb. What is the final concentration of DMSO? What is the medium used? What is the hyphal inducer?

In Results:
The first sentence is incomplete. What about the month of August?
In Figure 1: the legends are a little too small to read. The essential oil samples were combined or used in combination (to adapt in legend).
Line 239: Do the authors mean the MFC of the essential oil samples themselves or in combination with fluconazole (which seems to be described from Line 254)? What do the author mean by Fluconazole modulation? Same line 254: what is a potential modulatory action? The authors presumably suspected and tested the activity of the essential oil samples in combination with the commonly used antifungal therapeutic agent fluconazole.
The essential oil samples are seemingly non-cidal to the Candida species tested. Is it then still appropriate to mention the “fungicidal” concentration? The inhibitory concentration seems more suited here, hence the suggestion to adapt the method to an MIC test rather than MFC.
In the experiment where the essential oil samples were combined to fluconazole: was the viability of the cells tested? Fluconazole inhibits growth of candida but does not kill. Was any of the combination cidal to the cells? The authors suggest a synergism between the plant-derived samples and the azole compound: do they render fluconazole cidal?
Could the authors refer to published data on how to preform drug combination assays? here is an example that may help: https://www.ncbi.nlm.nih.gov/pmc/articles/PMC3067183/
These assays have been fairly well established and standardized procedures are available. It may help in the interpretation of the data and in comparative analyses.

In Figure 2: are the indicated concentrations correct? Is it 8.192 ug/ml or 8.192 mg.ml? same for the bottom right figure.
The images are clear. However, it is difficult to interpret as it is not clear what the hyphal inducer was? Was it high temperature? Fluconazole typically blocks hyphal growth but does not block germ tube formation in liquid cultures at least. Typically, a good inducer is serum, but I understand that the authors kept the same medium as their drug assays for comparative reasons. However, shouldn’t the highest concentration inhibit growth itself? Could other hyphal inducing conditions bet tested to make final conclusions. Lines 263-266 are very general, yet they can only be made in that particular growth condition.

---

## Round 0.2 · Minor Revisions

Please make the remaining minor revisions highlighted by reviewer 3.

Reviewer 1 ·

Basic reporting

the manuscript is in the journal's scope and meets the standards of the journal

Experimental design

All methods and techniques were sound and fully respected

Validity of the findings

the results are original and are highlighted soundly.

Additional comments

All remarks made previously were taken into account

Reviewer 3 ·

Basic reporting

no comment

Experimental design

no comment

Validity of the findings

no comment

Additional comments

The manuscript has been revised according to most comments made. Here are some minor comments, which are mainly typos:

In the abstract:
line.53: antifungal activity against three Candida species and that it can act in synergy with fluconazole would be clearer…please remove “would be cleare” which was part of my comment but should not be part of the sentence.

In intro:
line.100: is the first study to describe .to describe the chemical….please correct the typo here

In methods:
Line 111: It presents a Table 1 and receipt of an average of 1.043 mm (mm) of rainfall per year (FUNCEME, 2016), where they concentrate between January and May with a dry period that lasts between….it presents a Table 1 does not sound correct: adjust that whole sentence.

Line 122: Fresh leaves of the species Psidum salutare were collected from three individuals…..do you mean that three people collected the leafes? It does not seem like it is an information that is needed in method. Please adjust this sentence

Line 170: The antifungal fluconazole (from Sigma - F8929 ≥ 98% (HPLC), powder) it was diluted in sterile water (16.384 μg/mL), and oil were previously diluted in dimethyl sulfoxide (DMSO - Dynamic) and sterile distilled water was added in order to obtain the desired concentration for the tests (16384 μg/mL) both oil and fluconazole were posteriorly microdiluted in Sabouraud Dextrose Broth (SDB) medium in a serial concentration manner ranging from 8192 to 8 μg/mL in 96-well plates…..this sentence is a little too long and there are several typos in it.
Line 177: the fact that DMSO concentration was not constant through out the experiments may affect the results as DMSO alone at high concentration can affect cell viability. Did the authors test the effect of 5% DMSO on Candida cells in the same conditions as the ones used for the MIC assay for example?

Table 3 Title: “Modulation of the essential oil of salutare on Candida (ug/ml) e IC50 (ug.ml)”. What is the meaning of “modulation” here? Do you mean the effect? Add P. salutare, and not salutare. And is it IC50 and not e IC50?

---

## Round 0.3 · accepted · Accept

Thanks for dealing with those final comments. I look forward to seeing your paper in press!

#